# *Sulfolobus acidocaldarius* Microvesicles Exhibit Unusually Tight Packing Properties as Revealed by Optical Spectroscopy

**DOI:** 10.3390/ijms20215308

**Published:** 2019-10-25

**Authors:** Alexander Bonanno, Robert C. Blake, Parkson Lee-Gau Chong

**Affiliations:** 1Department of Medical Genetics and Molecular Biochemistry, The Lewis Katz School of Medicine at Temple University, Philadelphia, PA 19140, USA; tuf20683@temple.edu; 2College of Pharmacy, Xavier University of Louisiana, New Orleans, LA 70125, USA; rblake@xula.edu

**Keywords:** thermoacidophilic archaea, microvesicles, red edge excitation shift (REES), intrinsic protein fluorescence, Laurdan, glycerol dialkyl calditol tetraether (GDNT), glycerol dialkyl glycerol tetraether (GDGT), generalized polarization (GP), membrane probe, liposomes

## Abstract

In this study, we used optical spectroscopy to characterize the physical properties of microvesicles released from the thermoacidophilic archaeon *Sulfolobus acidocaldarius* (Sa-MVs). The most abundant proteins in Sa-MVs are the S-layer proteins, which self-assemble on the vesicle surface forming an array of crystalline structures. Lipids in Sa-MVs are exclusively bipolar tetraethers. We found that when excited at 275 nm, intrinsic protein fluorescence of Sa-MVs at 23 °C has an emission maximum at 303 nm (or 296 nm measured at 75 °C), which is unusually low for protein samples containing multiple tryptophans and tyrosines. In the presence of 10–11 mM of the surfactant n-tetradecyl-β-d-maltoside (TDM), Sa-MVs were disintegrated, the emission maximum of intrinsic protein fluorescence was shifted to 312 nm, and the excitation maximum was changed from 288 nm to 280.5 nm, in conjunction with a significant decrease (>2 times) in excitation band sharpness. These data suggest that most of the fluorescent amino acid residues in native Sa-MVs are in a tightly packed protein matrix and that the S-layer proteins may form J-aggregates. The membranes in Sa-MVs, as well as those of unilamellar vesicles (LUVs) made of the polar lipid fraction E (PLFE) tetraether lipids isolated from *S. acidocaldarius* (LUV_PLFE_), LUVs reconstituted from the tetraether lipids extracted from Sa-MVs (LUV_MV_) and LUVs made of the diester lipids, were investigated using the probe 6-dodecanoyl-2-dimethylaminonaphthalene (Laurdan). The generalized polarization (GP) values of Laurdan in tightly packed Sa-MVs, LUV_MV_, and LUV_PLFE_ were found to be much lower than those obtained from less tightly packed DPPC gel state, which echoes the previous finding that the GP values from tetraether lipid membranes cannot be directly compared with the GP values from diester lipid membranes, due to differences in probe disposition. Laurdan’s GP and red-edge excitation shift (REES) values in Sa-MVs and LUV_MV_ decrease with increasing temperature monotonically with no sign for lipid phase transition. Laurdan’s REES values are high (9.3–18.9 nm) in the tetraether lipid membrane systems (i.e., Sa-MVs, LUV_MV_ and LUV_PLFE_) and low (0.4–5.0 nm) in diester liposomes. The high REES and low GP values suggest that Laurdan in tetraether lipid membranes, especially in the membrane of Sa-MVs, is in a very motionally restricted environment, bound water molecules and the polar moieties in the tetraether lipid headgroups strongly interact with Laurdan’s excited state dipole moment, and “solvent” reorientation around Laurdan’s chromophore in tetraether lipid membranes occurs very slowly compared to Laurdan’s lifetime.

## 1. Introduction

Microvesicles (MVs) released from the thermoacidophilic archaeon *Sulfolobus acidocaldarius* (designated as Sa-MVs) contain considerably fewer proteins (29 [1]) than MVs released from bacteria and mammalian cells (hundreds [2,3]). The most abundant protein species in Sa-MVs are the S-layer proteins, which cover the surface of the microvesicles as well as the cells [1]. Two S-layer proteins, namely, SlaA (~45 kDa) and SlaB (~120 kDa) [4], were found in *S. acidocaldarius*. They self-assemble to form a regular array of pores (pore size: 2–8 nm [5]) with a p3 symmetry [6]. While the S-layer forms a tough barrier for the majority of the cell or microvesicle surface, small molecules can reach the underlying lipid membrane via the S-layer pores and may even permeate through the membrane entering the interior of the cells or microvesicles.

Lipids in Sa-MVs are exclusively tetraethers [1]; in contrast, lipids in MVs released from bacteria or mammalian cells are predominately diesters. In Sa-MVs, glycerol dialkyl glycerol tetraether (GDGT or caldarchaeol; Figure 1) is the dominating lipid hydrophobic core structure (99.8% by weight [1]), while glycerol trialkyl glycerol tetraether (GTGT, Figure 1) is a minor lipid component (0.2% by weight [1]). No glycerol dialkyl calditol tetraether (GDNT or calditolglycerocaldarchaeol; Figure 1) was found in Sa-MVs [1], despite that GDNT is the major lipid component (~70% (*w*/*w*) of the total isoprenoid ethers) of the *S. acidocaldarius* plasma membrane [7].

To date, very little is known (reviewed below) about the molecular structures and physical behaviors of Sa-MVs. Transmission electron microscopy (TEM) showed that Sa-MVs are largely spherical in the pH range 2.6–7.2 [1,8]. Dynamic light scattering (DLS) data revealed that Sa-MVs have a diameter ~180 nm under the cell’s optimum growth conditions (75–80 °C and pH 2–3) and that the particle size of Sa-MVs remains virtually unchanged in the temperature range 25–80 °C [8]. The size of Sa-MVs decreases by ~40 nm in response to a pH change from 2.6–3.0 to 4.3–7.2, probably due to the conformational changes of the S-layer proteins or partly due to the dissociation of Sa-MV aggregates [8]. The zeta potential of Sa-MVs in the growth media at pH 2.6 and 70 °C is about −0.5 mV, and the isoelectric point for Sa-MVs in 1 mM KCl is 3.0 [8].

In the present study, we used fluorescence spectroscopy, in conjunction with dynamic light scattering, absorption spectroscopy, and surfactant treatment, to further characterize the physical properties of Sa-MVs. We employed both the Sa-MV’s intrinsic protein fluorescence and the fluorescence from the extrinsic membrane probe 6-dodecanoyl-2-dimethylaminonaphthalene (Laurdan). Specifically, we used the emission and excitation/absorption spectra of Sa-MV proteins to assess the environment near the tryptophan and tyrosine residues and test if there exist H- or J-aggregates of protein chromophores. We also used the generalized polarization (GP) and the red edge excitation shift (REES) of Laurdan fluorescence to examine the solvent relaxation and motional restriction of the probe in Sa-MV membranes. The GP and REES values of Laurdan fluorescence in Sa-MVs are compared with those obtained from the unilamellar vesicles (LUVs) made of the polar lipid fraction E (PLFE) lipids isolated from *S. acidocaldarius* (LUV_PLFE_) and the LUVs reconstituted from the tetraether lipids extracted from Sa-MVs (LUV_MV_) as well as the LUVs made of the synthetic diester lipids. PLFE is composed exclusively of tetraether lipids GDNT (~90%) and GDGT (~10%) [9], whereas Sa-MV lipids are made of GDGT and GTGT. Our data allows us to evaluate whether this lipid compositional difference leads to differential membrane behaviors among Sa-MVs, LUV_PLFE_, and LUV_MV_. The results obtained from this study shed light on the dynamic structures and packing properties of Sa-MVs and pave the way for future studies on the structure–activity relationship of Sa-MVs and the development of Sa-MVs as naturally occurring nanoparticles for technological applications [8].

## 2. Results and Discussion

### 2.1. Intrinsic Protein Fluorescence

The emission spectrum of the intrinsic fluorescence of Sa-MVs in Tris buffer freshly isolated from the cell suspensions is presented in Figure 2 (black curve, labeled as “undialyzed”). Three emission peaks (Peaks I, II and III) were detected at 316, 373, and 420 nm, respectively, when excited at 275 nm. After the dialysis (MWCO 25 kDa, Spectra/Por membrane, Spectrum, Houston, TX) of Sa-MVs against the same Tris buffer (1 mL sample versus 2 L buffer) for 4 h, the intensities of Peaks II and III were reduced significantly, while the emission wavelengths of these two peaks did not shift (Figure 2, green curve). Additional dialysis against 2 L buffer with the buffer changed every four hours for 48 h further reduced the fluorescence intensities of Peak II and Peak III while Peak I remained as the dominating fluorescence peak of Sa-MVs (Figure 2, blue curve). The emission spectrum of the intrinsic Sa-MV fluorescence is virtually unchanged after 48 h of dialysis (Figure 2, red curve for 72 h dialysis).

Light at 275 nm excites both tryptophans and tyrosines in Sa-MVs. Based on the emission wavelengths at the peaks, it can be suggested that Peaks II and III in Figure 2 do not originate from tryptophans nor tyrosines in Sa-MV proteins. Since Peaks II and III can be significantly reduced to a minimal level by dialysis, these two fluorescence peaks most likely come from some small molecules, either originally produced by the cells or inherited from the cell growth environment, that are loosely bound to Sa-MVs. Extensive dialysis can reduce but cannot completely eliminate these two fluorescent peaks, suggesting that these small molecules also have tight binding to Sa-MV membranes or some Sa-MV proteins. The particle size and the PDI value of Sa-MVs do not change with dialysis (Appendix A), indicating that those small molecules that are removed by dialysis are not S-layer proteins nor membrane lipids. A thorough investigation of the chemical identifies of Peaks II and III will be undertaken in the future and is beyond the scope of the current study. In the present study, we focus on Peak 1 (Figure 2) because we believe it originates from protein intrinsic fluorescence, which would bear more biological interests such as the structures and dynamics of protein folding and association.

Extensive dialysis caused the apparent emission maximum of Peak I to shift from 316 nm to 303 nm (Figure 2, measured at room temperature, ~23 °C). Intrinsic protein fluorescence peaked around 303 nm is rare, which requires some discussion. Intrinsic protein fluorescence mainly comes from tryptophans and tyrosines. Since tyrosine has an extinction coefficient four times lower than tryptophan and since tyrosine fluorescence is often quenched in the protein by a variety of mechanisms including energy transfer to tryptophans, tryptophan usually dominates intrinsic protein fluorescence [11]. A prominent example is bovine serum albumin (BSA), which has 18 tyrosines and 2 tryptophans. While tryptophan takes up only 10% of the fluorescent aromatic amino acids (disregarding phenylalanine due to its low extinction coefficient and fluorescence quantum yield), the emission spectrum of BSA in solution is dominated by tryptophans, showing a maximum fluorescence intensity at 345 nm [12].

The fluorescence emission spectrum of dialyzed Sa-MVs shown in Figure 2 could be an exception, in addition to the case of the bacterial elongation factor Tu [13], to the general trend that tryptophan dominates intrinsic fluorescence of proteins that contain both tryptophans and tyrosines. Sa-MVs have 29 distinct proteins and most, if not all, Sa-MV proteins contain multiple tryptophans and tyrosines [1]. The most abundant protein species in Sa-MVs are the S-layer proteins [1], namely, SlaA and SlaB [4]. SlaA from *S. acidocaldarius* contains 96 tyrosines and 6 tryptophans [14]. To our knowledge, the amino acid composition of SlaB from *S. acidocaldarius* has not been reported yet, however, SlaB from a similar thermoacidophilic archaeon *Sulfolobus solfataricus* contains 16 tyrosines and one tryptophan [15]. Since about 95% of the fluorescent aromatic amino acids in the S-layer proteins are tyrosines and since the fluorescence of free tyrosine in solution at neutral pH has an emission maximum at 303 nm [16], it is not unreasonable to observe that the emission maximum of dialyzed Sa-MVs occurs at 303 nm when excited at 275 nm (Figure 2). It is possible that, unlike the case of BSA [12], tyrosine fluorescence in Sa-MVs is not quenched and tryptophan fluorescence is not dominating.

A tightly packed protein matrix in Sa-MVs near tyrosine and tryptophan residues could be another plausible explanation for the observation of the emission maximum at the low wavelength 303 nm. One precedent is the fluorescence of *Pseudomonas fluorescens* azurin, which has an emission maximum at 305–308 nm when excited at 275 nm [17]. *Pseudomonas fluorescens* azurin is a single tryptophan protein, and its low wavelength for emission maximum has been attributed to the apolar environment surrounding the tryptophan residue, which is almost completely shielded from water [18]. Another precedent is the fluorescence of the mutant dI component of transhydrogenases from *Rhodospir illum rubrum*, which gives an emission maximum at 303.5 nm due to the tryptophan residues embedded in a very rigid microenvironment [19]. It is possible that, like azurin and the dI mutant, most of the fluorescent amino acid residues in Sa-MVs are also in a tightly packed, water-shielded protein matrix, thus giving such an extreme low wavelength for the emission maximum. However, unlike azurin, which is molecularly dispersed in solution, S-layer proteins, the most dominant proteins in Sa-MVs, aggregate at the particle surface and are supposed to from regular crystalline lattices as they do on the surface of the archaea cells. In addition, some other proteins in Sa-MVs are also anchored in the tightly packed tetraether lipid membrane located under the S-layer. Thus, it is of interest to test if there is spectroscopic evidence for the chromophores of the fluorescent aromatic amino acid residues in Sa-MV proteins (mainly S-layer proteins) to from H- or J-aggregates [20].

We have used the surfactant n-tetradecyl-β-d-maltoside (TDM) as a tool to investigate the spectral difference in intrinsic protein fluorescence between native Sa-MVs (aggregation of S-layer proteins) and microvesicles completely disrupted by TDM (dis-aggregation of S-layer proteins). The critical micelle concentration (CMC) of TDM is 0.015 mM at 20 °C [21]. As shown in Figure 3 (top), the particle size of dialyzed Sa-MVs increases by ~100 nm from 0 to ~0.2 mM TDM, which probably reflects the insertion of the surfactant into the vesicle membrane, according to the three-stage model of the surfactant–membrane interactions [22,23]. At ~0.2 mM, TDM insertion reaches the limit. Thereafter, mixed micelles are formed, and vesicle disruption occurs, eventually leading to small fragments of 20–30 nm at high [TDM] (Figure 3, top). A similar trend was seen in our previous study of Sa-MVs against the surfactant Triton X-100 [8].

The bottom panel of Figure 3 shows that there are significant differences in spectral properties between Sa-MVs in the absence of TDM and presence of 10.13 mM TDM. In the absence of TDM, the emission maximum of Sa-MVs appears at 303.5 nm and the excitation maximum is at 288 nm. At (TDM) = 10.13 mM, the emission max appears at 312 nm and the excitation maximum is at 280.5 nm. There is a red shift by 7.5 nm in the excitation spectrum and an increase in excitation band sharpness, with the width of half maximum changed from 22.2 nm to 9.3 nm, when S-layer proteins are changed from the dis-aggregated state ((TDM) = 10.13 mM) to the aggregated state ((TDM) = 0) (Figure 3).

The red shift, in conjunction with an increase in excitation band sharpness, is indicative of the occurrence of J-aggregates [20,24,25]. In J-aggregates, molecules are aligned parallel, with the transition dipole moments of the chromophore coherently coupled, head-to-tail, resulting in delocalization of electronically excited states over many (typically 3–50) monomeric chromophores [25,26]. When forming J-aggregates, the excitation energy level is lowered and as a result the excitation/absorption band undergoes a red shift.

Absorption measurements of Sa-MVs (Figure 4) yielded the results consistent with those obtained from fluorescence measurements (Figure 2 and Figure 3). Figure 4 (curve a) shows that the major absorption band of un-dialyzed Sa-MVs peaked at 287 nm, with a broad shoulder in the region 310–390 nm. This broad shoulder is significantly reduced after 72 h of dialysis against buffer (curve b, Figure 4). The dialyzed Sa-MVs in the absence of TDM exhibit an absorbance maximum at 288 nm (curve b, Figure 4). In the presence of 11 mM TDM, when the S-layer proteins are dis-aggregated, the absorption maximum of dialyzed Sa-MVs is shifted to a shorter wavelength, 285 nm (curve c, Figure 4), similar to the trend observed in Figure 3. The excitation spectra shown in Figure 3 are sharp because they were determined by measuring the fluorescence intensity at 315 ± 8 nm, which monitored only the intrinsic protein fluorescence. In contrast, the protein absorption spectra shown in Figure 4 were relatively broad due to the presence of non-protein “impurities” (i.e., those responsible for Peaks 1 and 2 in Figure 2).

Figure 5A,B show that the emission maximum of Sa-MVs in Tris buffer decreases monotonically and significantly with increasing temperature, changing from 303–305 nm at 20 °C to 296 nm at 75.4 °C. This temperature-induced blue shift is reversible upon cooling. A similar blue shift in λ_em,max_ was observed (Appendix A) when Sa-MVs were dispersed in 50 mM pyrophosphate buffer (pH 7.2). This test excludes the possibility that the blue shift in λ_em,max_ shown in Figure 5B comes from the pH change due to the use of Tris buffer as the temperature dependence of pH is much less in pyrophosphate buffer than in Tris buffer.

This 7–9 nm blue shift in emission maximum (Figure 5B) is an interesting observation because elevated temperature usually unfolds the proteins, increases protein volume fluctuations, and makes aromatic amino acid residues more exposed to aqueous phase, which would lead to a red shift in emission maximum. Apparently, these did not happen to proteins in Sa-MVs. It is likely that the majority of the proteins in Sa-MVs (i.e., S-layer proteins) are retained in a very tightly packed situation over the entire temperature range examined (20–75 °C). This assertion is reasonable as liposomal membranes made of bipolar tetraether lipids have been shown to exhibit relatively low temperature dependence of solute permeation and volume fluctuation [27,28] and as S-layer proteins and many other proteins in Sa-MVs are membrane bound [1]. The observed blue shift in emission maximum with increasing temperature may simply arise from (i) thermal agitation that disorders the orientation of solvent dipoles with respect to the excited state dipole moments of protein chromophores and (ii) the thermal-induced increase in distance between solvent dipoles and the excited state dipole moments of protein chromophores, as described by Macgregor and Weber [29]. Unlike the emission maximum, the excitation maximum of dialyzed Sa-MVs does not vary with temperature (Figure 5D), probably due to the low strength of the ground state dipole moment of tryptophans and tyrosines. The steady decrease of the intensity of intrinsic protein fluorescence with increasing temperature (Figure 5C) is expected, as a result of increased dynamic quenching.

### 2.2. Laurdan Fluorescence

Laurdan and 6-propionyl-2-dimethylaminonaphthalene (Prodan) have the same chromophore, and the fluorescence of both probes is extremely sensitive to environmental polarity changes mainly due to the large change in the probe’s dipole moment when the probes are excited [30]. The high polarizability of the probes triggers varying degrees of solvent reorientation around each of the probe’s excited state “monopoles” [29]. In membrane studies, the term “solvent” mentioned above is referring to the bulk water molecules and the water molecules bound to the lipid polar headgroups as well as the polar residues of the lipids near the probe’s chromophore. Solvent relaxation in the bulk aqueous water is on the picosecond timescale whereas solvent relaxation due to bound water and nearby lipid polar headgroups occurs in nanoseconds [31]. Thus, when interpreting the fluorescence data of these two probes (fluorescence lifetime ~3 ns) from membrane studies, bound water and lipid polar headgroups around the chromophore are more important factors than bulk water.

In proteoliposomes, Prodan, which carries a 3-C propionyl tail, can insert into lipid membranes or bind non-covalently to proteins [32], whereas Laurdan, which carries a 12-C lauroyl chain, would only insert into lipid membranes. Thus, in this study, we chose to use Laurdan as an environmentally sensitive probe to explore the organization and dynamics in the interfacial regions between the lipid hydrophobic core and the lipid polar headgroups in Sa-MV membranes. The results are compared with various LUVs. LUV_DPPC_ and LUV_POPC_ are composed of the diester lipids DPPC and POPC, respectively, whereas lipids in Sa-MVs, LUV_MV_, and LUV_PLFE_ are exclusively tetraethers as mentioned earlier.

Laurdan can insert into diester lipid membranes with its chromophore located near the polar headgroup regions [33,34,35] and with its chromophore’s dipole moment aligned in parallel with the membrane normal [36]. When the membrane packing in diester lipid membranes is tight and the water content near the chromophore is low, the generalized polarization (GP) value of Laurdan fluorescence is high, and vice versa [37]. Our GP data on LUV_DPPC_ and LUV_POPC_ shown in Figure 6 are fully consistent with this well-established trend. Below the main phase transition temperature of DPPC (41 °C), the GP value is high (0.45 at 20 °C); above the transition temperature, GP becomes low (e.g., −0.1 at 50 °C) (Figure 6). The abrupt change in GP is centered around 41 °C (Figure 6), which reflects the DPPC main phase transition. In the temperature range examined, LUV_POPC_ is entirely in the liquid crystalline state; thus, as expected, there is no abrupt change in GP with temperature and the GP values are low compared to those in LUV_DPPC_ (Figure 6).

The disposition of Laurdan in tetraether lipid membranes is very different from that in diester lipid membranes, as revealed by our previous fluorescence microscopy study on PLFE and DPPC giant unilamellar vesicles (GUVs) [38]. The photoselection experiment showed that the dipole moment of Laurdan’s chromophore in GUV_PLFE_ is aligned in parallel with the membrane surface, whereas, in sharp contrast, the dipole moment of Laurdan’s chromophore in GUV_DPPC_ is aligned perpendicular to the membrane surface [38]. The most plausible configuration for Laurdan in GUV_PLFE_ is that the lauroyl tail of Laurdan inserts into the hydrocarbon core in parallel with the dibiphytanyl chains while the chromophore resides in the lipid polar headgroup regions with the long molecular axis of the chromophore aligned in parallel with the membrane surface [38]. This unusual L-shape disposition for Laurdan in PLFE liposomes is presumably caused by the rigid/tight packing in PLFE liposomes and by the steric hindrance of the branched methyl group in PLFE lipids [38]. In this disposition, the chromophore of Laurdan in GUV_PLFE_ is in close proximity to the bound water molecules at the polar headgroups and close to the lipid headgroup polar moieties (e.g., phosphoinositol and other sugar moieties, Figure 1), and as a result, the degree of “solvent” relaxation is always extensive—hence the low GPs (near zero) at all of the temperatures examined in our previous GUV study (12–66 °C) [38]. Because the chromophore disposition is different, Laurdan’s GP values obtained from tetraether lipid membranes cannot be directly compared with those obtained from diester lipid membranes.

Figure 6B shows that, when excited at 390 nm over the entire temperature range examined (18–68 °C), the GP values of LUV_PLFE_ are low (from 0 to −0.2). These data are in good agreement with the GP values (from 0 to −0.3 in the temperature range 12–66 °C) previously measured from GUV_PLFE_ using the same excitation wavelength (two photon excitation at 780 nm, which is equivalent to one photon excitation at 390 nm) [38]. When excited at 340 nm, similar results on the temperature dependence of Laurdan’s GP were obtained from LUV_PLFE_ (Figure 6A). In this case, the GP values are slightly higher (Figure 6A) due to the excitation of probe molecules in the less solvent relaxed state. Regardless of the excitation wavelength, the GP values of LUV_PLFE_ are much lower than the GP values of gel state DPPC at any given temperature (Figure 6). It would be totally wrong if these GP data were interpreted as membrane packing in LUV_PLFE_ being more loose than that in LUV_DPPC_ as our previous volume fluctuation measurements clearly showed that membrane packing in PLFE liposomes is much tighter than that in DPPC liposomes at any given temperature in the range 16–80 °C [28]. Thus, our present data (Figure 6) echo the previous finding that the GP values from tetraether lipid membranes cannot be directly compared with the GP values from diester lipid membranes. This point applies to GUVs as well as LUVs.

It is, however, meaningful to compare Laurdan’s GP values in Sa-MVs, LUV_MV_, and LUV_PLFE_ because lipids in these three membranes are exclusively tetraethers. The compositional differences among these three systems are: (1) Sa-MVs and LUV_MV_ have GDGT and GTGT as the hydrophobic cores whereas LUV_PLFE_ have GDNT and GDGT as the core structures (Figure 1); (2) the chemical structures of the polar headgroups in PLFE are known (see Figure 1), however, the structures of the polar headgroups in Sa-MV lipids have not been determined yet. There is a high possibility that the polar headgroups in Sa-MV lipids are very different from those in PLFE lipids; and (3) Sa-MV membrane is covered by an S-layer and contains membrane-bound proteins whereas LUV_MV_ and LUV_PLFE_ are protein free.

In these three tetraether lipid-based membrane systems, Laurdan’s GP decreases with increasing temperature monotonically without any abrupt changes. This result indicates that there is a continuous reduction in membrane packing tightness as the temperature increases and that there is no sign for lipid phase transitions. This is a piece of new information about Sa-MVs and the LUVs reconstituted from Sa-MV lipids, but a bit surprising with regards to lipid phase transitions in PLFE liposomes. GUV_PLFE_ (~10 μm) have previously been shown to exhibit a small phase transition at ~50 °C by using Laurdan’s GP [38]. The lack of an abrupt change in GP with temperature in LUV_PLFE_ (~200 nm) (Figure 6) may be attributed to subtle changes in membrane packing due to the increase in vesicle curvature.

To further assess membrane packing tightness and gain knowledge about the dynamic structure (e.g., solvent reorientation) near the probe’s chromophore, we have also calculated the red edge excitation shift (REES) of Laurdan fluorescence in Sa-MVs and various liposomes. The REES values are the differences in the wavelength of emission maximum when comparing the emission spectra excited at 340 nm (near the excitation maximum) with those excited at 390 nm (the red edge excitation) (see Materials and Methods). The interactions between a fluorophore and its surroundings in membranes could be quite complex, which would result in a series of microscopically inhomogeneous sub-states [39]. When excited at the red edge, only a small population of the sub-states will be excited. Those would be the ones that interact most strongly with the environment in the excited state and least strongly in the ground state [39]. If the interaction of fluorophore with its surroundings remains unchanged or changes little (e.g., solvent reorientation does not occur or it is extremely slow) during the fluorescence lifetime, the emission spectra when excited at the red edge will be shifted significantly to longer wavelengths (i.e., REES), compared to those excited at the absorption band maximum [39]. Therefore, generally speaking, probe in a motionally more restricted environment will exhibit a higher REES value.

Figure 7 illustrates how the emission spectrum of Laurdan in the membrane systems examined changes upon the red edge excitation at different temperatures. It is clear from Figure 7 that there is a large red shift in the emission spectrum of Laurdan in Sa-MVs, LUV_MV_, and LUV_PLFE_ when excited at 390 nm (a red edge excitation). These three membrane systems contain exclusively tetraether lipids. In contrast, a red shift is not immediately obvious for diester liposomes, i.e., LUV_DPPC_ and LUV_POPC_, by simple visual examination of the emission spectra (Figure 7).

In Figure 8, the calculated REES values are plotted against temperature for all the membrane systems examined (see Appendix A for the actual REES values). The data indicate that the REES values can be divided into two groups. In the tetraether lipid membrane systems (i.e., Sa-MVs, LUV_MV_ and LUV_PLFE_), the REES values are high (9.3–18.9 nm). In diester lipid membranes (i.e., LUV_POPC_ and LUV_DPPC_), the REES values are relatively low (0.4–5.0 nm). These data suggest that Laurdan in tetraether lipid membranes is in a motionally much more restricted environment than Laurdan in diester liposomes at any given temperature examined [40,41]. The high REES (Figure 8) and low GP (Figure 6) values of Laurdan fluorescence in tetraether lipid-based membrane systems suggest that the bound water molecules and the polar lipid moieties in the polar headgroup regions of tetraether lipid membranes, where the chromophore of Laurdan resides, strongly interact with Laurdan’s excited-state dipole moment [29] and that the solvent reorientation around the chromophore occurs very slowly compared to Laurdan’s fluorescence lifetime [42].

Among these three tetraether membrane systems examined, the REES values of Sa-MVs (14.9–18.9 nm) are significantly higher than those in LUV_MV_ (10.2–14.0 nm) and LUV_PLFE_ (9.3–12.6 nm) (Appendix A), with a *p* value less than 0.003 comparing Sa-MVs to LUV_PLFE_. A plausible explanation is that the crystalline array of the S-layer proteins on top of the tetraether lipid membrane in Sa-MVs provides additional motional restriction to Laurdan’s chromophore.

It is also worthy of noting that REES in LUV_MV_ is higher than that in LUV_PLFE_, especially at low temperatures (<50 °C, *p* < 0.03 (*n* = 3)) (Figure 8) and that Sa-MVs and LUV_MV_ have higher GP values than LUV_PLFE_ (Figure 6). These data suggest that Laurdan’s chromophore experiences tighter membrane packing when Laurdan is in Sa-MVs and LUV_MV_ than in LUV_PLFE_. This packing difference may originate from the subtle difference in lipid composition. As discussed earlier, PLFE has GDNT and GDGT as the hydrophobic cores whereas Sa-MV lipids have GDGT and GTGT (Figure 1). All these three lipids are bipolar tetraether macrocyclic or semi-macrocyclic molecules. GDNT has calditol on one side and glycerol on the other side of the hydrocarbon core whereas GDGT and GTGT have glycerol at both polar ends. In addition, the polar headgroups on Sa-MV lipids, which are not yet known, may be very different from those seen in PLFE. These differences in chemical structure may lead to changes in hydrogen bonding between Laurdan and lipids and among lipids themselves, all of which can affect membrane packing and solvent reorientation.

Figure 8 shows that there is no abrupt change in REES with temperature in Sa-MVs, LUV_MV_ LUV_PLFE_ and LUV_POPC_, which indicates that there is no major phase transition in those membrane systems in the temperature range examined. These results are consistent with the GP data (Figure 6). In these membranes, the slight decrease in REES with increasing temperature (Figure 8) likely results from the thermal-induced increase in volume fluctuations and consequently a slight increase in the rate of solvent reorientation.

REES in LUV_DPPC_ undergoes an abrupt change at 41 °C (Figure 8), which reflects the main phase transition of DPPC. This result is in excellent agreement with the GP data (Figure 6). However, it is surprising that the REES value is high in DPPC liquid crystalline (fluid) state and low in gel state (Figure 8). Intuitively, this trend would be the other way around. One possible explanation is that REES not only depends on the rate of solvent reorientation around the probe, but also changes with probe location and orientation in the membrane. A recent quantum mechanical and molecular dynamics study [35] showed that the location and orientation of Laurdan’s chromophore in DPPC bilayers have a heterogeneous distribution, rather than a single fixed value, and that the fluorescence properties of Laurdan in lipid bilayers are influenced by the chromophore’s orientation and depth of membrane penetration. It is possible that, through the DPPC gel-to-fluid phase transition, the Laurdan’s depth penetration and orientation are changed resulting in an increase in REES, despite that solvent reorientation increases somewhat with increasing temperature.

## 3. Materials and Methods

### 3.1. Growth of Archaea and Isolation of Sa-MVs

*S. acidocaldarius* cells (strain DSM639; ATCC) were grown aerobically and heterotrophically at ~75 °C and pH ~2.6, as described [10]. Sa-MVs were isolated using ultrafiltration (MWCO = 100 kDa, Millipore, Burlington, MA, USA) and a series of centrifugations on the cell suspensions as described [1,8]. The procedures used in our lab to isolate Sa-MVs are summarized in Appendix A.

### 3.2. Formation of Liposomes Using the Total Lipids Extracted from Sa-MVs and the PLFE Lipids Isolated from S. acidocaldarius

PLFE lipids were isolated from *S. acidocaldarius* cells as previously published [9]. Sa-MV lipids were extracted using a solvent mixture of chloroform, methanol and water (34.5:34.5:31, *v*/*v*/*v*) [8]. Stock solutions of Sa-MV lipids and PLFE lipids were prepared using the same solvent mixture. Aliquots of the lipid stock solutions were pipetted out, mixed, and dried first by nitrogen gas and then lyophilization overnight. The dried lipid film was then rehydrated with a buffer solution (either 50 mM Tris containing 10 mM EDTA and 0.02% sodium azide, pH 7.2 or 50 mM sodium pyrophosphate adjusted to pH 7.2 with 1 M citric acid) and vortexed at ~70 °C to make multilamellar vesicles (MLVs). The MLVs were then extruded (Lipex Biomembranes, Vancouver, Canada) 10 times through two 200-nm Nucleopore polycarbonate membranes at ~70 °C to create large unilamellar vesicles (LUVs). LUVs of the diester lipids 1-palmitoyl-2-oleoyl-*sn*-glycero-3-phosphocholine (POPC) and dipalmitoyl-*sn*-glycero-3-phosphocholine (DPPC) (Avanti Polar Lipids, Alabaster, AL, USA) were generated by the same extrusion method at ~50 °C. LUVs made of PLFE, Sa-MV lipids, DPPC and POPC are abbreviated as LUV_PLFE_, LUV_MV_, LUV_DPPC_ and LUV_POPC_, respectively. The particle size and polydispersity index (PDI) of Sa-MVs and LUVs were determined by dynamic light scattering on a Malvern Zetasizer 1000HS spectrometer (Wores, UK).

### 3.3. Chemicals and Reagents

Stock solution (70 µM) of Laurdan (Avanti Polar Lipids, Alabaster, AL, USA) was made in *N,N′*-dimethylformamide (DMF, Sigma-Aldrich, St. Louis, MO, USA). Stock solution (40–80 mM) of n-tetradecyl-β-d-maltoside (TDM, Sigma-Aldrich, St. Louis, MO, USA) was prepared in deionized water. The concentrations of POPC, DMPC, and PLFE stock solutions were determined by the method of Bartlett [43]. The concentration of Sa-MVs is expressed in terms of the total protein content as determined by the Lowry assay [44].

### 3.4. Spectroscopy Measurements

Emission and excitation spectra were measured using an ISS K2 fluorometer (Champaign, IL, USA). Temperature of the samples in the cuvette was controlled using a circulating water bath and measured by a thermal couple. Excitation spectra of intrinsic protein fluorescence were measured using a ratio mode based on the emission intensity monitored at 315 nm through an emission monochromator (slit width = 4 nm) over the intensity measured from rhodamine B in ethylene glycol (3 mg/mL) plus a RG630 filter in the reference channel. Technical emission spectra (as opposed to the corrected emission spectra) of intrinsic protein fluorescence were measured using the excitation light at 275 nm with the slit width in both emission and excitation monochromators set at 8 nm. For the extrinsic membrane probe studies, Sa-MVs, LUV_PLFE_ and LUV_MV_ were incubated with aliquots of Laurdan (in DMF) with stirring at ~65 °C for 2 h, whereas LUV_DPPC_ and LUV_POPC_ were doped with Laurdan with stirring at ~50 °C for 1 h. The probe-to-lipid molar ratio was ~1/400 for LUV_PLFE_, LUV_DPPC_ and LUV_POPC_. The probe content in Sa-MVs was 0.29 nmol of Laurdan per μg of protein. The technical emission spectra of Laurdan in Sa-MVs and various liposomes were recorded from 370 nm to 600 nm or from 408 nm to 600 nm using excitation at 340 nm or 390 nm, respectively. The background readings from Sa-MVs or liposomes without the probe were subtracted from the sample readings. All the emission spectra presented in this study are technical emission spectra.

The excitation GP was calculated from the resulting emission spectra using the equation: GP = (I_440_ − I_490_)/(I_440_ + I_490_). Here I_490_ and I_440_ are the fluorescence intensities at 490 and 440 nm, respectively. To obtain the Laurdan’s red edge excitation shift (REES) value, the wavelength-based emission spectra were converted to the wave number-based spectra. The REES value was determined by subtracting the center of mass of Laurdan’s wave number-based emission spectra excited at 340 nm from that excited at 390 nm.

Absorption spectra were measured on an OLIS CLARiTY 1000A spectrophotometer (OnLine Instrument Systems, Bogart, GA, USA) as described [45]. This instrument is a novel integrating cavity absorption meter able to conduct accurate absorbance measurements in particle suspensions (e.g., Sa-MVs) that scatter light. Identical 8-mL solutions of 50 mM Tris buffer with 10 mM EDTA and 0.02% (*w*/*v*) sodium azide were added to both the sample and reference observation cavities of the spectrophotometer. After recording a stable baseline, a volume was withdrawn from the sample cavity and replaced with an equal volume of suspended microvesicles. The contents of both observation cavities were maintained at 25 °C using a model TC-1 Peltier temperature control element from Quantum Northwest (Liberty Lake, WA, USA). Raw absorbance spectra were collected at a rate of 6.2 scans over the wavelength range per second for two minutes, and the resulting spectra were averaged. These averaged raw absorbance values were subsequently converted to equivalent absorbance values per cm using Fry’s method [45] with analysis software provided by OnLine Instrument Systems.

## 4. Conclusions

We have used intrinsic protein fluorescence/absorption and Laurdan fluorescence to study dynamic structures and packing properties of microvesicles released from *S. acidocaldarius* (Sa-MVs) over a wide range of temperatures (18–20 to 65–75 °C). The emission maximum of Sa-MV intrinsic protein fluorescence appears at 296–303 nm, which is rare, one of the bluest ever reported, and could result from tight packing in the microvesicles due to, for example, protein J-aggregate formation in the S-layer. Laurdan’s GP values in Sa-MVs are low, probably due to the special L-shaped disposition of Laurdan in tetraether lipid membranes in response to tight/rigid membrane packing as previously proposed [38]. The REES effect of Laurdan is most pronounced in Sa-MVs among all the membranes examined, suggesting that “solvent” reorientation around Laurdan’s chromophore in Sa-MVs occurs very slowly due to tight packing. These spectroscopic measurements provide consistent data in support of the proposition that Sa-MVs are unusually tightly packed. This level of understanding may help reveal the structure–activity relationship of Sa-MVs [46] and pave the way for developing Sa-MVs into a useful nanoparticle [8].

## Figures and Tables

**Figure 1 ijms-20-05308-f001:**
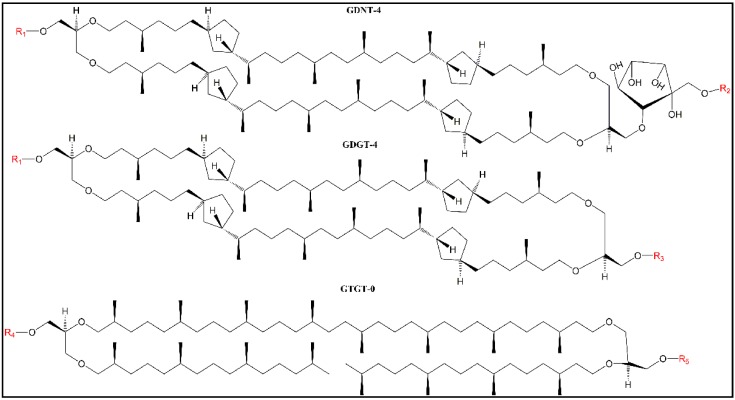
Illustration of the hydrophobic core structures (in black) of tetraether lipids found in the plasma membrane (glycerol dialkyl calditol tetraether (GDNT), glycerol dialkyl glycerol tetraether (GDGT), and glycerol trialkyl glycerol tetraether (GTGT)) [7] and microvesicles (GDGT and GTGT) [1]) of *Sulfolobus acidocaldarius* (Sa-MVs). R1–R5 (in red) are the hydrophilic headgroups. For polar lipid fraction E (PLFE) isolated from *S. acidocaldarius*, R1 = phospho-*myo*-inositol, R2 = β-d-glucopyranose, and R3 = β-d-galactopyranosyl- β-d-glucopyranose [10], and no GTGT was reported to be present. In the case of Sa-MVs, GDNT is not found and the chemical structures of R1, R3, R4, and R5 are not known [1]. The number after the abbreviations indicates the number of cyclopentane rings per molecule. In the cases of GDNT and GDGT isolated from *S. acidocaldarius*, this number can vary from 0–8.

**Figure 2 ijms-20-05308-f002:**
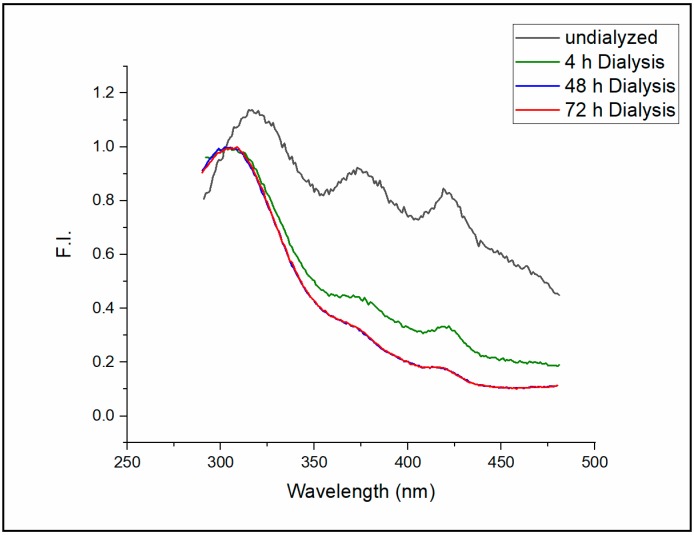
Effect of dialysis on the technical emission spectra of Sa-MVs isolated from *S. acidocaldarius* cell suspensions. λ_ex_ = 275 nm; slit width = 8 nm for both the excitation and the emission monochromator; temperature = 23 °C.

**Figure 3 ijms-20-05308-f003:**
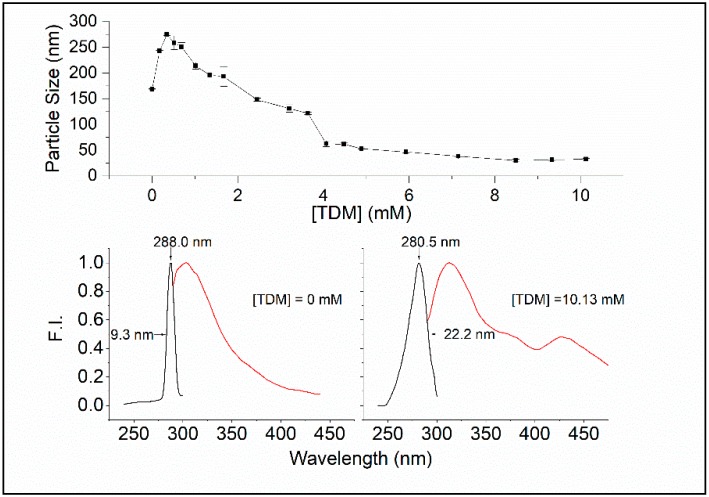
(top) The effect of n-tetradecyl-β-d-maltoside (TDM) on the particle size of Sa-MVs in 50 mM Tris containing 10 mM ethylenediaminetetraacetic acid (EDTA) and 0.02% NaN_3_ (pH 7.2). Temperature = 23 °C. (bottom) The excitation (black) and emission (red) spectrum of Sa-MVs in the absence of TDM (left) and in the presence of 10.13 mM TDM (right). The samples contained 36.8 micrograms of Sa-MV proteins/mL. Fluorescence intensities of the excitation spectra were recorded at 315 nm through an emission monochromator with a 4-nm slit width, and the exciting light was selected via an excitation monochromator with an 8-nm slit width. For the emission spectra, the excitation was fixed at 275 nm and the emission was measured through a monochromator with an 8-nm slit width.

**Figure 4 ijms-20-05308-f004:**
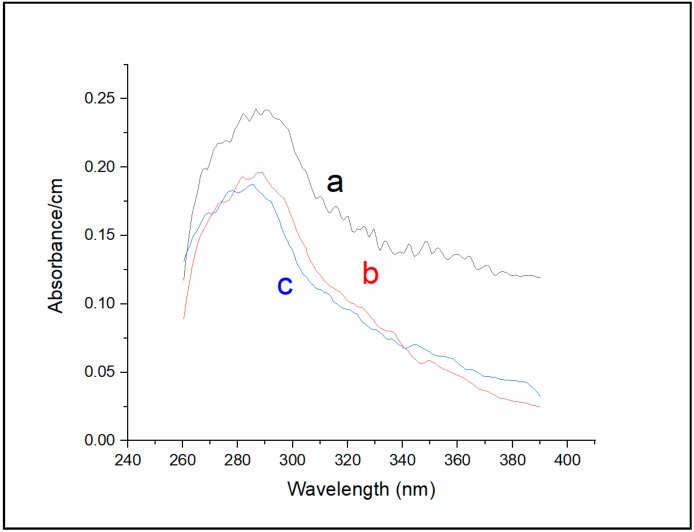
Absorption spectra of undialyzed Sa-MVs (**a**), Sa-MVs after 72 h dialysis (**b**), and dialyzed Sa-MVs in the presence of 11 mM TDM (**c**), measured at room temperature.

**Figure 5 ijms-20-05308-f005:**
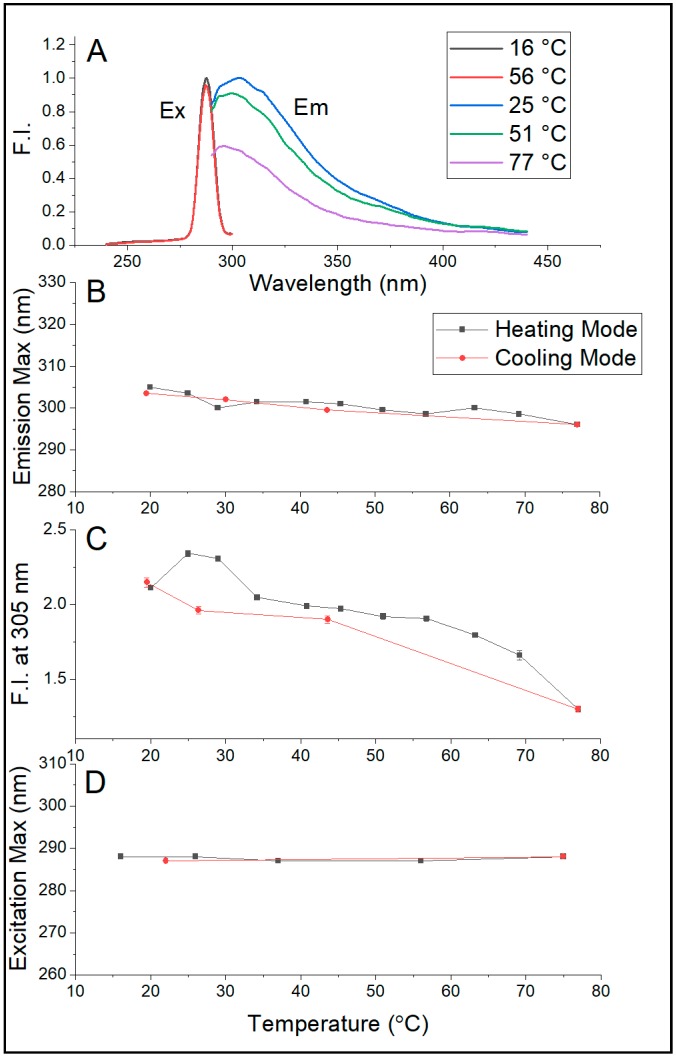
Effect of temperature on the intrinsic protein fluorescence of dialyzed Sa-MVs. (**A**) illustration of the temperature effect on the emission and excitation spectra; (**B**–**D**, respectively) the temperature dependence of the wavelength of the emission maximum (λ_em,max_), the fluorescence intensity (F.I.) measured at 305 nm when excited at 275 nm, and the wavelength of the excitation maximum (λ_ex,max_). Buffer: 50 mM Tris containing 10 mM EDTA and 0.02% sodium azide, pH 7.2. Error bars are standard deviations of three measurements.

**Figure 6 ijms-20-05308-f006:**
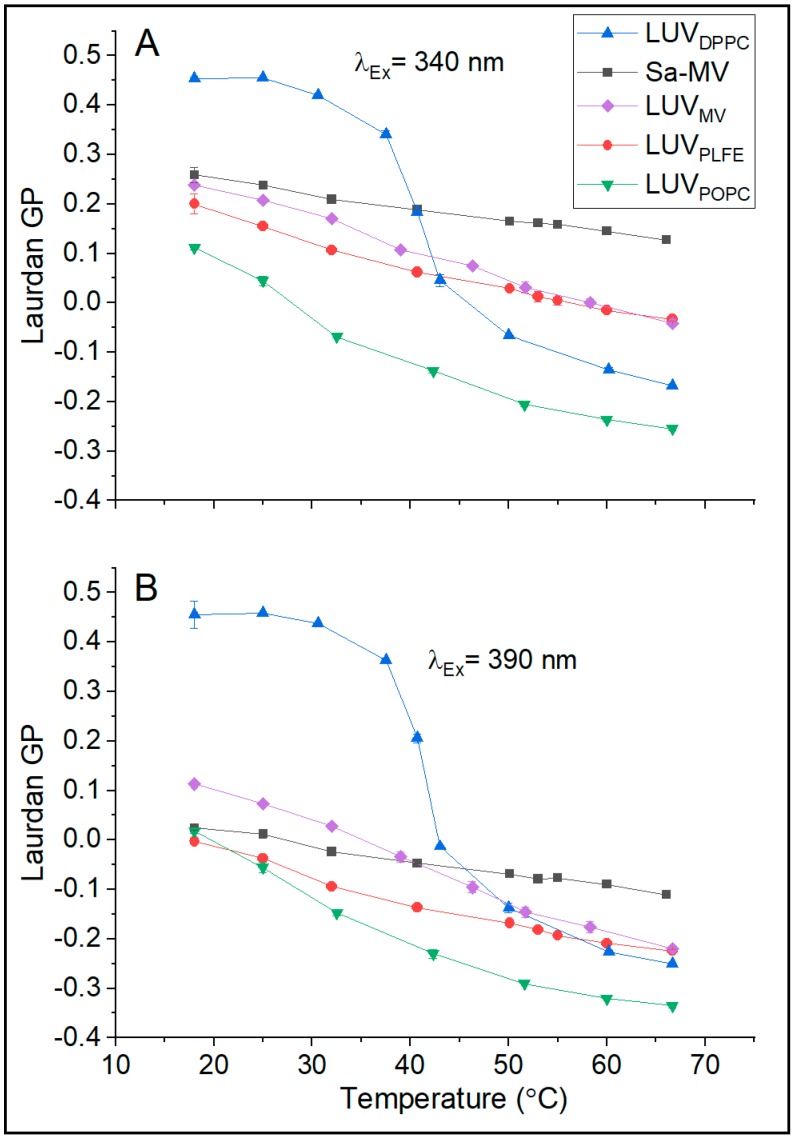
Effects of temperature on Laurdan’s generalized polarization (GP) in Sa-MVs and in various liposomes as indicated. (**A**) λ_ex_ = 340 nm; (**B**) λ_ex_ = 390 nm. All the samples were in the same Tris buffer described earlier. Error bars are the standard deviations of three measurements. The particle size and polydispersity index (PDI) values of Sa-MVs and various liposomes used in the Laurdan fluorescence measurements are given in Appendix A.

**Figure 7 ijms-20-05308-f007:**
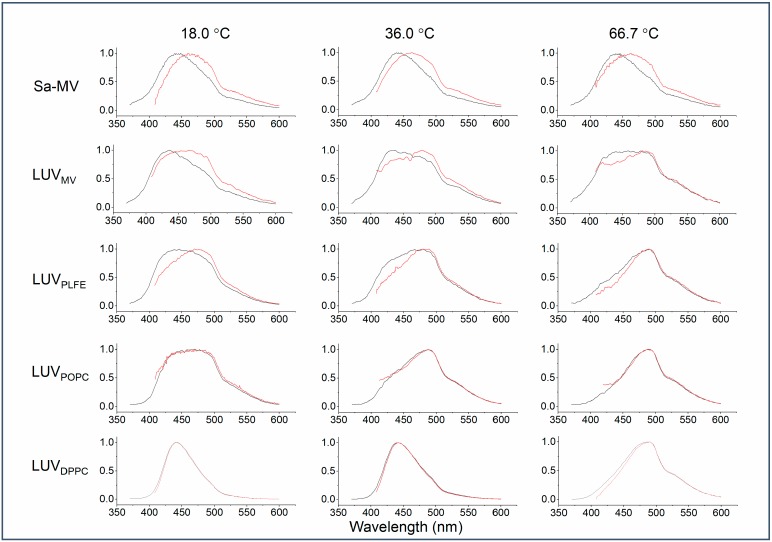
Illustration of technical emission spectra of Laurdan fluorescence in Sa-MVs and various liposomes using excitation wavelength = 340 nm (black) and 390 nm (red).

**Figure 8 ijms-20-05308-f008:**
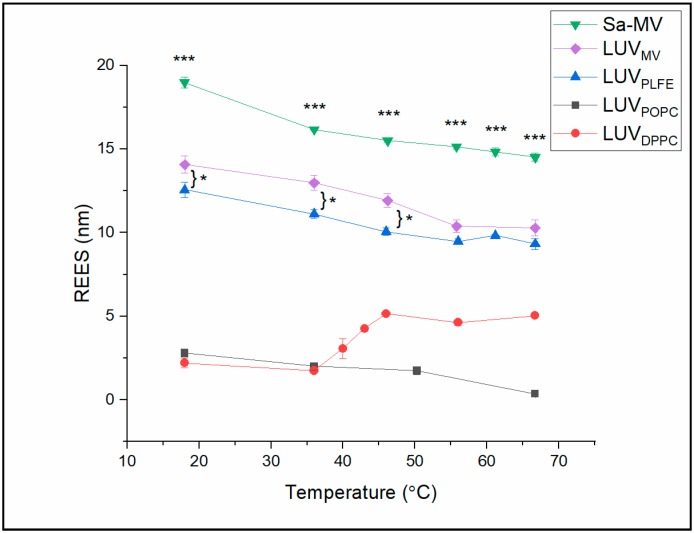
Effect of temperature on red edge excitation shift (REES) in Sa-MVs and various liposomes. Error bars are the standard deviations of three measurements. * denotes *p* value < 0.03 comparing large unilamellar vesicles (LUVs) reconstituted from the tetraether lipids extracted from Sa-MVs (LUV_MV_) to LUVs made of the polar lipid fraction E (PLFE) lipids isolated from *S. acidocaldarius* (LUV_PLFE_) and *** denotes *p* value < 0.003 comparing Sa-MV to LUV_PLFE_. The *p* values were calculated using a Student’s paired t-test.

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
