# Peer review of "Sulfolobus acidocaldarius Microvesicles Exhibit Unusually Tight Packing Properties as Revealed by Optical Spectroscopy"

_ijms, 2019, doi:10.3390/ijms20215308_

Round 1

Reviewer 1 Report

The manuscript presented by Bonanno et al. presents spectroscopic data on the behaviour of fluorescence electronic transitions of fluorophores in Sulfolobus acidocaldarius micro vesicles.  The authors examine the intrinsic excitation and emission spectra of the micro vesicles and they also present data for the membrane-interacting fluorophore Laurdan. The authors measured parameters sensitive to motion of the fluorophores and their molecular environment.

 All of the data are consistent with a system (i.e. the micro vesicles) that experiences extremely little molecular motion during the ~3 nm fluorescent lifetime of the micro vesicles or the interacting Laurdan. As the authors point out the wavelength of maximum intrinsic fluorescence is almost unprecedentedly blue shifted from that of other tryptophan-containing proteins.  The generalized polarization data from the Laurdan fluorescence also indicate very restricted motion.  

These to lines of evidence are consistent with the conclusion of the manuscript, namely that molecular motion is quite constrained over the time period of several nanoseconds, that is to say, the system exhibits very "tight packing properties."

This Reviewer found the manuscript clear and well written.  There are only a handful of small typographical errors to correct.  Otherwise, the manuscript is suitable for publication in This Reviewer's opinion.

This Reviewer is curious about the fluorescence properties of a denatured system.  Do they look more similar to "normal" fluorescence?  Is there a reason why the authors did not do this?

Author Response

We would like to thank this reviewer for the comment that this manuscript is clear and well written and suitable for publication.

Reviewer's comment: “This Reviewer is curious about the fluorescence properties of a denatured system. Do they look more similar to "normal" fluorescence? Is there a reason why the authors did not do this?”

Response: It would be interesting to compare the results from native Sa-MVs with those from denatured Sa-MVs. In fact, the TDM (detergent) data shown in Figure 3 has already partly addressed this issue. We observed a significant difference in optical properties between native Sa-MVs and detergent-treated Sa-MVs. In the presence of 11 mM TDM, many of the proteins in Sa-MVs are presumably “denatured” to some extent. However, it is not clear how to obtain temperature-denatured Sa-MVs because Sa-MVs appear to be resistant to high temperature (ref #8). In short, at present it is rather difficult to obtain a well-defined denatured Sa-MV system for comparison. However, definitely, this is an interesting subject for our future studies.

Reviewer’s comment: “There are only a handful of small typographical errors to correct.”

Response:  We have endeavored to correct typographical errors in the revised manuscript.

Reviewer 2 Report

Bonanno et al. report about optical spectroscopy to characterize the physical properties of microvesicles released from the thermoacidophilic archaeon Sulfolobus acidocaldarius (Sa-MVs).

This is a good manuscript and balanced assessment of the status of vesicle surface forming an array of crystalline structures. The article highlights important data that might have been overlooked when promulgating the scientific value of Optical Spectroscopy and related studies.

Only 2 areas need revision:

Page 14, figure 8: the notion p value <0.03 and <0.003 that these data from convey is critical and the message needs an explicit sentence or two at end of paragraph or in materials and methods regarding the statistics used. Are those data normally distributed (gaussian distribution). If this is not the case a non-parametric test should be used. If this is the case, a clear statement would be required. Page 15, Conclusion: the assertion consistent data in support of the proposition perhaps would require  to be rephrased to indicate that the value and sense of doing these experiments is open to question, with attendant translational implications, or softer wording to that effect.

Author Response

We appreciate this reviewer’s comment that “this is a good manuscript and balanced assessment of the status of vesicle surface forming an array of crystalline structures” and that “the article highlights important data that might have been overlooked when promulgating the scientific value of Optical Spectroscopy and related studies.”

Reviewer’s comment:  “Page 14, figure 8: the notion p value <0.03 and <0.003 that these data from convey is critical and the message needs an explicit sentence or two at end of paragraph or in materials and methods regarding the statistics used. Are those data normally distributed (gaussian distribution). If this is not the case a non-parametric test should be used. If this is the case, a clear statement would be required.”

Response: We have added a sentence (the p values were calculated using a Student's paired t-test) in the legend to Figure 8 to indicate how the p values were calculated.

Reviewer’s comment:  “Page 15, Conclusion: the assertion consistent data in support of the proposition perhaps would require  to be rephrased to indicate that the value and sense of doing these experiments is open to question, with attendant translational implications, or softer wording to that effect.”

Response:  We have rewritten the Conclusion Section.

In addition, we have endeavored to correct typographical errors throughout the entire manuscript.